# Are *Enterobacteriaceae* and *Enterococcus* Isolated from Powdered Infant Formula a Hazard for Infants? A Genomic Analysis

**DOI:** 10.3390/foods11223556

**Published:** 2022-11-08

**Authors:** Julio Parra-Flores, Adriana Cabal-Rosel, Beatriz Daza-Prieto, Pamela Chavarria, Eduard Maury-Sintjago, Alejandra Rodriguez-Fernández, Sergio Acuña, Werner Ruppitsch

**Affiliations:** 1Department of Nutrition and Public Health, Universidad del Bío-Bío, Chillán 3800708, Chile; 2Austrian Agency for Health and Food Safety, Institute for Medical Microbiology and Hygiene, 1220 Vienna, Austria; 3Department of Food Engineering, Universidad del Bío-Bío, Chillán 3800708, Chile

**Keywords:** powdered infant formula, *Enterobacteriaceae*, infants, antibiotic resistance, virulence

## Abstract

Powdered infant formulas (PIF) are the most used dietary substitutes that are used in order to supplement breastfeeding. However, PIF are not sterile and can be contaminated with different microorganisms. The objective of this study was to genomically characterize *Enterobacteriaceae* (ENT) and *Enterococcus* strains that were isolated from PIF. Strains were identified by matrix-assisted laser desorption ionization–time-of-flight mass spectrometry (MALDI-TOF MS) and whole-genome sequencing (WGS). Genomic typing, detection of virulence, and resistance profiles and genes were performed with the Ridom SeqSphere+ software; the comprehensive antibiotic resistance database (CARD) platform; ResFinder and PlasmidFinder tools; and by the disk diffusion method. Nineteen isolates from PIF were analyzed, including ENT such as *Kosakonia cowanii*, *Enterobacter hormaechei*, *Franconibacter helveticus*, *Mixta calida*, and lactic acid bacteria such as *Enterococcus faecium*. The strains exhibited resistance to beta-lactams, cephalosporins, and macrolides. Resistance genes such as *AcrAB-TolC*, *marA*, *msbA*, *knpEF*, *oqxAB*, *fosA*, *bla_ACT-_*_7_, *bla_ACT-_*_14,_
*qacJ*, *oqxAB*_,_
*aac(6’)-Ii*, and *msr(C)*; and virulence genes such as *astA*, *cheB*, *cheR*, *ompA ompX*, *terC*, *ironA*, *acm*, and *efaAfm*, *adem* were also detected. All the analyzed strains possessed genes that produced heat-shock proteins, such as IbpA and ClpL. In PIF, the presence of ENT and *Enterococcus* that are multiresistant to antibiotics—together with resistance and virulence genes—pose a health risk for infants consuming these food products.

## 1. Introduction

Powdered milk formulas intended for infants < 6 mo (PIF) are the most used dietary substitutes that are used in order to supplement breastfeeding; however, these products are not sterile [1,2]. For this reason, food safety control is a fundamental objective of regulatory agencies and manufacturers as this product is consumed by infants whose immune system is vulnerable and whose intestinal flora is still developing [3]. 

In 2004, after reviewing the published peer-reviewed literature on the relationship of bacterial infections associated with PIF, the Food and Agriculture Organization of the United Nations and the World Health Organization (FAO–WHO) decided to include the different microorganisms that contaminate PIF as disease risks. These infectious microorganisms were classified into three disease risk groups: Category A: clear evidence of causality for *Salmonella* and *E. sakazakii* (*Cronobacter* spp.); Category B: plausible but not yet proven causality for *Enterobacter vulneris*, *Citrobacter koseri*, *Enterobacter cloacae*, *Hafnia alvei*, *Pantoea agglomerans*, *Klebsiella pneumoniae*, and *K. oxytoca*; and Category C: less plausible or not yet proven causality for *Clostridium botulinum*, *Staphylococcus aureus*, *Listeria monocytogenes*, and *Bacillus cereus* [4].

Although only *Salmonella* and *Cronobacter* spp. are currently identified as important agents of neonatal infection through contaminated PIF, the emergence of other species of the *Enterobacteriaceae* (ENT) genus, as opportunistic infection hazards, have recently gained relevance due to their association with several reports of disease outbreaks and cases. A recent Center for Disease Control (CDC) study, which summarized data collected from 2011 to 2017, found that ENT caused between 23% and 31% of infections in adult, pediatric, and oncology wards [5]. Therefore, a zero tolerance approach has been defined for ENT in PIF [6].

In addition, several authors have reported the presence of different ENT such as *Franconibacter helveticus*, *Enterobacter hormaechei*, *Kosakonia cowanii*, and *Enterobacter cloacae* in the various PIF that are consumed by infants [7,8]. There is also the risk of infection caused by these opportunistic pathogens in premature infants and newborns when they are identified in reconstituted PIF and enteral feeding tubes in hospitals [9,10]. 

*Enterococcus* is a Gram-negative bacteria closely related to the dairy industry due to the fact that they are used as starter cultures in cheese production processes. However, they are also associated with poor hygiene processes as these microorganisms live as part of the natural flora in the intestinal tract of animals and humans [11].

These microorganisms are classified as opportunistic pathogens, and the severity of their infection is associated with resistance to the antibiotics used to treat them. It has been mentioned that the *Enterobacter* genus has a natural resistance to ampicillin and first-generation cephalosporins. Antibiotic resistant genes such as *fosA*, *marA*, *bla_ACT_*, *oqx*, *tetA*, and *tetB* are also present [12,13]. In addition, the presence and expression of several virulence factors promote an even poorer prognosis, such as outer membrane proteins (OmpA and OmpX), heavy metal resistant operons, iron acquisition, exotoxins, and occurrence of fimbriae [14].

The use of whole-genome sequencing (WGS) has facilitated the in-depth study of pathogenic organisms by generating extensive information that helps to determine relationships and taxonomic differences between them. It is not only used in order to identify isolates, but also for comprehensive profiling and genotyping using conventional 7-loci multilocus sequence typing (MLST), core genome MLST (cgMLST), and/or single nucleotide poly-morphism (SNP) analysis; as well as molecular serotyping and the detection of genes associated with antibiotic resistance and virulence. More precise epidemiological links can be established as a result [15,16]. Therefore, the analysis of the complete genomes and their comparison enables a more complete analysis of the pathogenesis process of many more pathogens. 

Given that the presence of ENT and *Enterococcus* have been mentioned as indicators of post-process contamination in PIF, and that its impact on the safety of PIF has not been assessed, the objective of this study was to genomically characterize the *Enterobacteriaceae* and *Enterococcus* strains that were isolated from PIF.

## 2. Materials and Methods

### 2.1. Strains Used in This Study

This study included the analysis of 17 strains of *Enterobacteriaceae* (ENT) and 2 strains of lactic acid bacteria (LAB) that were isolated from 155 samples of powdered infant formula (PIF) produced in Mexico (75 samples) and Chile (80 samples). These isolates were collected from individual cans of PIF, which were purchased on a monthly basis in supermarkets and pharmacies; further, the monthly period was based on the regular restocking of products. This enabled greater variability in the origin of the PIF production batches.

### 2.2. Isolation and Primary Identification

The NCh 2676 (2002) isolation regulation was used for ENT. For each sample, 225 mL buffered peptone water (BPW, MERCK, Darmstadt, Germany) was added to 25 g PIF, homogenized in a stomacher at a mean velocity for 60 s, and incubated at 37 °C for 24 ± 2 h. Subsequently, 1 mL of the sample was placed on violet red bile glucose agar (VRBD, MERCK, Darmstadt, Germany) and incubated at 37 °C for 24 ± 2 h. Characteristic colonies with a precipitating halo were selected and confirmed by oxidase reaction. For LAB, 0.1 mL aliquots were inoculated on de Man, Rogosa, and Sharpe agar (MRS, MERCK, Darmstadt, Germany) and incubated at 30 ± 1 °C for 72 ± 3 h under aerobic conditions. After incubation, five typical LAB colonies (spherical shape and whitish color) were selected. These colonies were subjected to Gram staining and catalase reaction. 

All bacteria were primarily identified by matrix-assisted laser desorption ionization–time-of-flight mass spectrometry (MALDI-TOF MS) (Bruker, Billerica, MA, USA) using the MBT Compass IVD software 4.1.60 (Bruker).

### 2.3. Whole-Genome Sequencing (WGS) and Molecular Identification

Before whole-genome sequencing (WGS), all ENT and *Enterococcus* strains were cultured on Columbia blood agar plates (bioMérieux, Marcy-l’Étoile, France) at 37 °C for 24 h. The genomic DNA was isolated from bacterial cultures with the MagAttract HMW DNA Kit (Qiagen, Hilden, Germany) according to the manufacturer’s instructions. The amount of DNA was quantified on a Lunatic instrument (Unchained Labs, Pleasanton, CA, USA). Nextera XT chemistry (Illumina Inc., San Diego, CA, USA) was used to prepare the sequencing libraries for a 2 × 300 bp paired-end sequencing run on an Illumina MiSeq sequencer. Samples were sequenced to achieve a minimum of 80-fold coverage using standard Illumina protocols. The resulting FASTQ files were quality trimmed and de novo assembled with the SPAdes v. 3.9.0 software. Contigs were filtered for a minimum of 5-fold coverage and 200 bp minimum length with the Ridom SeqSphere+ software v. 8.0 (Ridom, Münster, Germany) [17]. A gene-by-gene genome-wide comparison was performed for the bacterial typing with the MLST+ task template function of SeqSphere+, which was previously described. In addition, the sequences of the seven housekeeping genes of the conventional multilocus sequence typing (MLST) scheme were extracted and analyzed with the Ridom SeqSphere+ software v. 8.0 (Ridom, Münster, Germany) [17]; further, the classical sequence types (ST) were assigned in silico. This taxonomic identification was supplemented with ribosomal MLST (rMLST) using 53 genes from the database at https://pubmlst.org/ (accessed on 11 August 2022) species-id according to Jolley et al. [18].

### 2.4. Antibiotic Resistance Profile 

The disk diffusion method was used in accordance with the recommendations of the Clinical and Laboratory Standards Institute (CLSI) [19]. The commercial disks were composed of ampicillin (10 μg), amikacin (30 µg), cephalothin (30 μg), chloramphenicol (30 μg), ceftriaxone (30 µg), cefotaxime (30 μg), cefepime (30 µg), gentamicin (10 μg), levofloxacin (5 μg), netilmicin (30 µg), oxacillin (1 µg), and sulfamethoxazole-trimethoprim (1.25/23.75 µg). The characterization of the resistance/susceptibility profiles was determined according to the CLSI guidelines. The *Escherichia coli* ATCC 25922 and *Pseudomonas aeruginosa* ATCC 27853 strains were used as references.

### 2.5. Detection of Antibiotic Resistance and Virulence Genes

Genomes were screened for virulence, in addition, antimicrobial resistance genes were detected using the virulence factor database (VFDB) [20] and AMRFinderPlus [21] integrated in SeqSphere+—the ResFinder tool from the Center of Genomic Epidemiology (CGE) (http://www.genomicepidemiology.org (accessed on 4 August 2022) and the comprehensive antibiotic resistance database (CARD) [22]. Thresholds for the target scanning procedure were set with a required identity of ≥90% to the reference sequence and an aligned reference sequence ≥99%. The comprehensive antibiotic resistance database (CARD) was used with the “perfect” and “strict” default settings for the purposes of sequence analysis; further, the AMRFinderPlus 3.2.3 task template available in Ridom SeqSphere+ v. 8.0 software (Ridom, Münster, Germany) using the EXACT method at 100% was utilized, as well as the BLAST alignment for protein identification available in the AMRFinderPlus database.

### 2.6. Detection of Plasmids 

The PlasmidFinder 2.1 tool was used to detect plasmids. The selected minimum identity was 95% (http://www.genomicepidemiology.org/ accessed on 4 August 2022) [23].

## 3. Results

### 3.1. Isolation and Primary Species Identification of Enterobacteriaceae (ENT) Isolates

Of the 17 ENT strains presumptively identified, 7 were confirmed by MALDI-TOF as *Kosakonia cowanii*, 5 as *Enterobacter hormaechei*, 2 as *Enterobacter cloacae*, 2 as *Franconibacter helveticus*, and 1 as *Mixta calida*. In addition, the two isolated LAB strains were identified as *Enterococcus faecium*. All strains were isolated from different samples.

### 3.2. Molecular Typing of Isolates

When using rMLST to supplement identification, the *E. cloacae* strains were determined as *K. cowanii*. The conventional MLST scheme determined that the *Franconibacter helveticus* strains were ST345; moreover, the *Enterobacter hormaechei* strains were ST817 and ST818, and the *Enterococcus faecium* strains were ST1533 and ST 2193 (Table 1).

### 3.3. Antibiotic Resistance Profiles

All the *E. hormaechei* strains were resistant to cephalothin and only two isolates were resistant to ampicillin (510176 and 510428). For *K. cowanii*, the 510179 strain was resistant to cephalothin and ampicillin, while the 510420 strain was resistant to ceftazidime and cephalothin. The two *F. helveticus* strains were resistant to ampicillin and 510439 was resistant to amoxicillin–clavulanic acid. The *Mixta calida* strain showed sensitivity to all the tested antibiotics. The two E. facecium strains were resistant to ceftazidime, gentamicin, and ampicillin. The 510421 strain was also resistant to ciprofloxacin (Table 2).

### 3.4. Detection of Antibiotic Resistance and Virulence Genes

Only 9 of the 17 evaluated ENT strains (53%) showed antibiotic resistant genes. All the strains of these nine isolates exhibited the *marA* gene associated with antibiotic efflux and reduced permeability to antibiotics. In addition, 88% (8/9) strains had gene encoding AcrAB-TolC associated with antibiotic target alteration and antibiotic efflux mechanism. Both *E. faecium* strains showed the *aac(6′)-Ii* and *msr(C)* resistant genes, which are associated with resistance to aminoglycosides and macrolides through antibiotic efflux and inactivation mechanisms (Table 3).

Virulence genes were found in 100% of the evaluated strains. The most prevalent genes were *astA*, *cheB*, *cheR*, *ompA*, and *ompX*, which were found in all ENT isolates. The *flgB* and *flgK* genes were detected in 94% (16/17) of the strains and the *ibpA* gene in 100%; moreover, these genes were associated with flagellum synthesis and heat-shock proteins, respectively. In addition, three of the five *E. hormaechei* strains showed the *terC* and *ironA* genes that are related to tellurite reduction and iron acquisition, respectively. All five *E. hormaechei* strains carried plasmids (Table 4). The *K. cowanii* strain 510420 exhibited three plasmids. The two *E. faecium* strains exhibited the same virulence gene profile with adesines, heat proteins, and the same type of plasmids (Table 4).

## 4. Discussions

The production of powdered milk for infants (PIF) has a low water content. However, various microorganisms can survive and adapt to environments with low water content, such as those in PIF production plants [24]. Although the heat treatment used in PIF is usually sufficient to inactivate the pathogens transmitted by food, PIF recontamination can occur due to the presence of thermoduric bacteria in the environment and production equipment [25].

The objective of the Codex Alimentarius is to provide practical guidance and recommendations to governments, industry, and health care professionals/caregivers of infants and young children—as appropriate—on the hygienic manufacture of PIF and on the subsequent hygienic preparation, handling, and use of reconstituted formulas. It recommends the absence of ENT in 10 g of PIF as a control indicator of adequate post-processing hygiene [6]. Therefore, the absence of ENT, when there is no associated risk of disease, provides additional protection for newborns. This is especially the case for premature, immunocompromised, and low (<2500 g) and very low (<1500 g) birth weight newborns during the rehydration process and administration of infant feeding with PIF [26]. *Enterococcus* bacteria are also an indicator of fecal contamination in food and can be used as a post-processing hygiene criterion for various foods and drinking water [27].

In our study, *E. hormaechei* and *K. cowanii* were the most prevalent microorganisms among the analyzed ENT. *Enterobacter hormaechei* has proven to be clinically significant due to its association with outbreaks of sepsis in neonatal intensive care units in several countries. It is also frequently isolated from infant food products, including PIF, with recent information highlighting its resistance to antibiotics [28,29]. Clinical isolates of *E. hormaechei* with different degrees of beta-lactam resistance have also been increasingly reported, as well as with the production of extended-spectrum β-lactamases, AmpC β-lactamases, and carbapenemases worldwide [30]. Of the five *E. hormaechei* isolates in our study, three were resistant to cephalothin and two strains to ampicillin. Furthermore, AmpC β-lactamases were found, especially *bla_ACT-7_* and *bla_ACT-14_*, and there was a relationship between phenotype and resistance genes. These types of AmpC *β-lactamases* have also been reported in Brazil, the United States, and China [31,32,33]. Some 80% of our *E. hormaechei* isolates exhibited the *AcrAB-TolC* efflux pump system associated with multiple resistance to drugs, which have been reported in *E. coli*, *Salmonella*, and other ENT strains [34,35]. As for virulence genes, all the *E. hormaechei* strains showed the *astA* gene encoding of enteroaggregative *E. coli* heat-stable enterotoxin 1 (EAST-1) [36] that has been associated with severe outbreaks of diarrhea in humans [37]. 

*Kosaconia cowanii*, formerly known as *Enterobacter cowanii*, is a microbial species that is recognized as a plant pathogen rather than a human pathogen [38]. Given its ability to live in different environments and under varying conditions, the bacterium is highly environmentally competitive; it has an enormous metabolic potential and is identified as a component of biofilm-forming communities [39]. Although reports of human infections are scarce, cases have been recently reported involving bacteremia in premature newborns and adults [40,41]. In 2020, a study reported the presence of *E. hormaechei* and *K. cowanii* in the PIF consumed throughout the Americas by the sequencing of the fusA gene in order to identify them [8]. Of the nine *K. cowanii* strains found in our study, one strain was resistant to cephalothin and ampicillin, while another was resistant to ceftazidime. Both strains showed the same resistance gene profile and the *AcrAB-TolC* efflux pump system associated with multiple antibiotic resistance was highlighted [34]. Both also exhibited an operon of multiple drug resistance, such as the *marA* and *fosA* genes whose mechanism is antibiotic inactivation and is present in approximately 80% of ENT [42]. All strains exhibited several virulence genes, especially the *ibpA* gene encoding heat-shock proteins that function as a protein support and maintain cellular proteostasis under stress conditions. Therefore, under heat treatment conditions, the *ibpA* gene acts as the first line of defense against irreversible and acute protein aggregation [43]. 

*Enterococcus* bacteria are ubiquitous microorganisms that are also found as normal flora in raw milk and fermented products. However, their presence in food is still a matter of debate due to their high prevalence as an agent in nosocomial infections [44]. Although studies related to the presence and load of *Enterococcus* in consumer-ready pasteurized milk are quite scarce, they are almost nonexistent in other foods such as PIF. In an Australian study, a prevalence of 20.8% positivity of *Enterococcus* was found in pasteurized milk, while in a more recent study with the same product it reached 82.9%. According to species, 0% and 27.8% *E. faecium* was found in these studies, respectively [45]. In our study, the two *E. faecium* strains showed the same phenotypic resistance profile, and the 510421 strain exhibited resistance to ciprofloxacin. Strains with antibiotic resistance to ciprofloxacin and gentamicin have been detected in pasteurized milk with values of 80.2% and 70.7%, respectively, as well as in non-pasteurized dairy products [46,47]. Therefore, if we consider that in addition to the phenotypic antibiotic resistance, i.e., the two *E. faecium* strains analyzed in our study that exhibited several virulence genes, the risk associated with the consumption of these food products by infants <6 mo of age is a wake-up call that we cannot ignore. It should be noted that the presence of the ClpL protein produced by class III heat-shock genes was found in both strains. They are associated with the cellular response in the presence of heat and osmotic stress; they are also found in several Gram-positive strains, such as heat-resistant *Listeria monocytogenes* [48]. The presence of these chaperones may contribute to their resistance to heat treatments, and thus to the presence of these microorganisms in finished and consumer-ready products. 

Therefore, these findings should be analyzed in terms of the risk associated with PIF consumption by infants and the lack of available scientific information. In addition, there is no adequate control of the processes carried out by PIF manufacturers and a lack of supervision by health authorities responsible for authorizing the commercialization of secure and safe PIF [49]. 

Since 2007, the World Health Organization (WHO) recommends using water at >70 °C to rehydrate PIF. Once rehydrated, if PIF is not immediately consumed, it must be refrigerated at <4 °C for no more than 24 h or discarded within 2 h if not consumed [50]. It is known that PIF is an excellent medium for the growth of microorganisms, including ENT [51].

## 5. Conclusions

The presence of *Enterobacteriaceae* and *Enterococcus* that are multiresistant to antibiotics—including resistance and virulence genes—in powdered infant formulas (PIF) manufactured in Chile and Mexico for distribution in the Americas are a health risk for infants who consume these products. It is important to emphasize that the Codex Alimentarius establishes a zero tolerance approach for *Enterobacteriaceae* in PIF. Therefore, the findings of our study are a wake-up call for authorities and manufacturers to improve production quality standards in order to prevent recontamination and to, therefore, produce safe PIF.

## Figures and Tables

**Table 1 foods-11-03556-t001:** Identification of the *Enterobacteriaceae* and *Enterococcus* species found in the study.

ID	Origin	Species	ST
*Enterobacteriaceae*			
510176	Mexico	*Enterobacter hormaechei*	818
510179	Mexico	*Kosakonia cowanii*	ND
510180	Mexico	*Kosakonia cowanii*	ND
510181	Chile	*Enterobacter hormaechei*	817
510375	Chile	*Enterobacter hormaechei*	818
510376	Mexico	*Enterobacter hormaechei*	818
510377	Chile	*Kosaconia cowanii*	ND
510378	Chile	*Kosakonia cowanii*	ND
510416	Chile	*Kosakonia cowanii*	ND
510418	Mexico	*Kosakonia cowanii*	ND
510420	Chile	*Kosaconia cowanii*	ND
510424	Chile	*Mixta calida*	ND
510428	Mexico	*Enterobacter hormaechei*	817
510430	Chile	*Kosaconia cowanii*	ND
510432	Chile	*Kosaconia cowanii*	ND
510439	Chile	*Franconibacter helveticus*	345
510440	Chile	*Franconibacter helveticus*	345
*Enterococcus*			
510417	Chile	*Enterococcus faecium*	1533
510421	Chile	*Enterococcus faecium*	2193

ID: Identification; ST: sequence type; and ND: not determined.

**Table 2 foods-11-03556-t002:** Antibiotic resistance profiles of the *Enterobacteriaceae* and *Enterococcus* strains.

Strains	CAZ (30 µg)	CTX (30 µg)	AMC (20/10 µg)	CIP (5 µg)	KF (30 µg)	W (30 µg)	GE (10 µg)	TE (30 µg)	CL (30 µg)	AM (10 µg)
*Enterobacteriaceae*										
510176	S	S	S	S	**R**	S	S	S	S	**R**
510179	S	S	S	S	**R**	S	S	S	S	**R**
510180	S	S	S	S	S	S	S	S	S	S
510181	S	S	S	S	**R**	S	S	S	S	S
510375	S	S	S	S	**R**	S	S	S	S	S
510376	S	S	S	S	**R**	S	S	S	S	S
510377	S	S	S	S	S	S	S	S	S	S
510378	S	S	S	S	S	S	S	S	S	S
510416	S	S	S	S	S	S	S	S	S	S
510418	S	S	S	S	S	S	S	S	S	S
510420	**R**	S	S	S	**R**	S	S	S	S	S
510424	S	S	S	S	S	S	S	S	S	S
510428	S	S	S	S	**R**	S	S	S	S	**R**
510430	S	S	S	S	S	S	S	S	S	S
510432	S	S	S	S	S	S	S	S	S	S
510439	S	S	**R**	S	S	S	S	S	S	**R**
510440	S	S	S	S	S	S	S	S	S	**R**
*Enterococcus*										
510417	**R**	S	S	S	S	S	**R**	S	S	**R**
510421	**R**	S	S	**R**	S	S	**R**	S	S	**R**

CAZ: Ceftazidime; CTX: cefotaxime; AMC: amoxicillin–clavulanic acid; CIP: ciprofloxacin; KF: cephalothin; W: nalidixic acid; GE: gentamicin; TE: tetracycline; CL: chloramphenicol; AM: ampicillin; R: resistance; and S: susceptibility.

**Table 3 foods-11-03556-t003:** Antibiotic resistant gene profiles of the *Enterobacteriaceae* and *Enterococcus* strains.

ID Strains	Species	Resistance Profile	Resistance Genes
*Enterobacteriaceae*			
510176	*E. hormaechei*	KF, AM	*AcrAB-TolC*, *marA*, *msbA*, *kpnEF*, *oqxAB*, *bla_ACT-_*_7 *(AmpC)*_
510179	*K. cowanii*	KF, AM	*AcrAB-TolC*, *marA*, *msbA*, *kpnEF*, *fosA*
510181	*E. hormaechei*	KF	*AcrAB-TolC*, *marA*, *msbA*, *knpEF*, *oqxAB*, *fosA*, *bla_ACT-_*_14*(AmpC)*_
510375	*E. hormaechei*	KF	*AcrAB-TolC*, *marA*, *msbA*, *knpEF*, *oqxAB*, *bla_ACT-_*_7 *(AmpC)*_
510376	*E. hormaechei*	KF	*AcrAB-TolC*, *marA*, *msbA*, *knpEF*, *oqxAB*, *bla_ACT-_*_7 *(AmpC)*_
510420	*K. cowanii*	CAZ	*AcrAB-TolC*, *marA*, *msbA*, *knpEF*, *fosA*
510428	*E. hormaechei*	KF, AM	*marA*, *msbA*, *knpE*, *fosA*, *oqxAB*, *qacJ*, *fosA*, *bla_ACT-14(AmpC)_*
510439	*F. helveticus*	AMC, AM	*AcrAB-TolC*, *marA*, *kpnF*, *qacJ*, *fosA*
510440	*F. helveticus*	AM	*AcrAB-TolC*, *marA*, *fosA*
*Enterococcus*			
510417	*E. faecium*	CAZ, GE, AM	*aac(6′)-Ii*, *msr(C)*
510421	*E. faecium*	CAZ, CIP, GE, AM	*aac(6′)-Ii*, *msr(C)*

ID strains: Identification strains; CAZ: ceftazidime; AMC: amoxicillin–clavulanic acid; CIP: ciprofloxacin; KF: cephalothin; GE: gentamicin; and AM: ampicillin. *AcrAB-TolC*: fluoroquinolone, cephalosporin, glycylcycline, penam, tetracycline, rifamycin, phenicol, disinfecting agents, and antiseptics; *marA*: fluoroquinolone, monobactam, carbapenem, cephalosporin, glycylcycline, cephamycin, penam, tetracycline, rifamycin, phenicol, penem, disinfecting agents, and antiseptics; *fosA:* fosfomicyn; *msbA*: nitroimidazole antibiotic; *kpnEF*: macrolide, aminoglycoside, cephalosporin, tetracycline, rifamycin, disinfecting agents, and antiseptics; *qacJ*: disinfecting agents and antiseptics; *oqxAB*: fluoroquinolone, glycylcycline, tetracycline, diaminopyrimidine, and nitrofuran antibiotics; *blaACT-44*(AmpC): betalactamase; *aac(6′)-Ii*: aminoglycoside; and *msr(C):* macrolide.

**Table 4 foods-11-03556-t004:** Presence of virulence genes and plasmids within the strains under study.

ID.	Species	Virulence Genes	Plasmids
*Enterobacteriaceae*			
510176, 510376	*E. hormaechei*	*terC*, *ironN*, *astA*, *cheB*, *cheR*, *flgB*, *flgK*, *ibpA*, *ompA*, *ompX*,	IncFIB(pECLA), IncFii(pECLA)
510179, 510180, 510377, 510378	*K. cowanii*	*astA*, *cheB*, *cheR*, *fic*, *flgB*, *flgK*, *ibpA*, *ompA*, *ompX*,	ND
510181	*E. hormaechei*	*astA*, *cheB*, *cheR*, *fic*, *flgB*, *flgK*, *ibpA*, *ompA*, *ompX*	Col(pHAD28), INcFIB(K), IncFii(Yp)
510375	*E. hormaechei*	*terC*, *ironN*, *astA*, *cheB*, *cheR*, *flgB*, *flgK*, *ibpA*, *ompA*, *ompX*	IncFIB(pECLA), IncFii(pECLA)
510416, 510418	*K. cowanii*	*astA*, *cheB*, *cheR*, *fic*, *flgB*, *flgK*, *ibpA*, *ompA*, *ompX*	Col440I
510420	*K. cowanii*	*astA*, *cheB*, *cheR*, *fic*, *flgB*, *flgK*, *ibpA*, *ompA*, *ompX*	Col440I; Col(MGD2); Col(pHAD28)
510424	*Mixta calida*	*astA*, *cheB*, *cheR*, *fic*, *flgB*, *ibpA*, *ompA*, *ompX*	IncFIB(K); IncHI1B
510428	*E. hormaechei*	*astA*, *cheB*, *cheR*, *fic*, *flgB*, *flgK*, *ibpA*, *ompA*, *ompX*	IncFII(Yp); IncFIB(K)
510430	*K. cowanii*	*astA*, *cheB*, *cheR*, *fic*, *flgB*, *flgK*, *ibpA*, *ompA*, *ompX*	Col440I
510432	*K. cowanii*	*astA*, *cheB*, *cheR*, *fic*, *flgB*, *flgK*, *ibpA*, *ompA*, *ompX*	ND
510439, 510440	*F. helveticus*	*astA*, *cheB*, *cheR*, *fic*, *flgB*, *flgK*, *ibpA*, *ompA*, *ompX*	ND
*Enterococcus*			
510417	*E. faecium*	*acm*, *scm*, *efaAfm*, ClpL	Rep3(rep11c; rep18a); Inc18(rep1); RepA_N(repUS15)
510421	*E. faecium*	*acm*, *scm*, *efaAfm*, ClpL	Rep3(rep11c; rep18a); Inc18(rep1; rep2); RepA_N(repUS15)

ID: Identification; *terC*: tellurium resistance; *ironN*: enterobactin siderophore receptor protein; *astA:* EAST-1 heat-stable toxin; *fic*: cell division; *flgB*: motility; *flgK*: flagellar hook-associated protein 1; *cheB*: desiccation tolerance; *cheR*: chemotaxis protein methyltransferase; *ibpA*: small heat shock protein; *ompA*: adhesion cell, biofilm formation; *ompX*: adhesion cell; *acm*: collagen adhesin precursor Acm; *scm*: collagen adhesin protein Scm; *efaAfm*: surface adhesin genes; and ClpL: heat-shock protein. ND: not detected.

## Data Availability

The *Enterobacteriaceae* isolates were submitted to https://pubmlst.org/organisms/cronobacter-spp (accessed on 31 August 2022) with the designation of ID 3664-3670. The two *Enterococcus* isolates were submitted to the international database at https://pubmlst.org/organisms/enterococcus-faecium (accessed on 29 August 2022) with the designations of ID 4909 and 4622. The data presented in this study are available on request from the corresponding author.

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
