# Peer review of "Are Enterobacteriaceae and Enterococcus Isolated from Powdered Infant Formula a Hazard for Infants? A Genomic Analysis"

_foods, 2022, doi:10.3390/foods11223556_

Round 1

Reviewer 1 Report

I want to thank you for the opportunity to participate in the manuscript review.

The introduction to the article is clearly written, it introduces the topic well, the selection of literature is appropriate. The scope of the article was correctly defined. The methodology is clearly written and makes it possible to repeat the experiments. The techniques used are modern and properly selected. The results and discussion are well presented. Tables are legible and properly signed. The literature is up to date.

I consider the entire manuscript interesting and worthy of attention. The manuscript was pleasant to read and I am impressed with the research done.

 Below are some small remarks, mainly from the Methodology, which do not affect my positive assessment.

 Line 91, 94 - Please specify the manufacturer, city and country of the microbial media that were used for the test.

Line 93 - How long was the incubation of the homogenized PIF?

Line 96 - Why were MRS agar plates incubated at 25°C? This is not a typical LAB temperature. According to the manufacturer's specifications, the plates should be incubated at 30°C for 72 h under aerobic conditions. Please explain.

Line 150, 161, 162, 165, 166, 177 - Please write species names in italics.

Line 198 - Please complete the period at the end of the sentence (after "table 4").

 Author Response

Point 1: The introduction to the article is clearly written, it introduces the topic well, the selection of literature is appropriate. The scope of the article was correctly defined. The methodology is clearly written and makes it possible to repeat the experiments. The techniques used are modern and properly selected. The results and discussion are well presented. Tables are legible and properly signed. The literature is up to date.

I consider the entire manuscript interesting and worthy of attention. The manuscript was pleasant to read and I am impressed with the research done.

Response Point 1: Thank you very much for your comments.

Point 2: Below are some small remarks, mainly from the Methodology, which do not affect my positive assessment.

Line 91, 94 - Please specify the manufacturer, city and country of the microbial media that were used for the test.

Response Point 2: This information was added to the manuscript.

Point 3: Line 93 - How long was the incubation of the homogenized PIF?

Response Point 3: This information was added to the manuscript.

Point 4: Line 96 - Why were MRS agar plates incubated at 25°C? This is not a typical LAB temperature. According to the manufacturer's specifications, the plates should be incubated at 30°C for 72 h under aerobic conditions. Please explain.

Response Point 4: This was a transcription error. We followed the manufacturer’s instructions, which establish 72 ± 3 hours at 30 ± 1 °C. This information was added to the manuscript.

Point 5: Line 150, 161, 162, 165, 166, 177 - Please write species names in italics.

Response Point 5: This was corrected in the manuscript.

Point 6: Line 198 - Please complete the period at the end of the sentence (after "table 4").

Response Point 6: The period was added.

Reviewer 2 Report

In this manuscript the authors describe what kind of bacterial contamination they have found in PIF from two countries, Mexico and Chile.

Although they have found contaminations, it is hard to determine the significance of the findings as it is unclear how many samples were tested to get to these numbers of contaminating strains. This needs to be included or at least discussed. Moreover it is also unclear if maybe some of the contaminations come from one sample.

E. hormaechei was found before in PIF. Was K. cowanii found before in PIF? This needs to be stated to determine the novelty. Also for resistance genes found the novelty (for PIF) needs to be stated. In general it is unclear if the results presented are novel or a confirmation of what was already known. For instances, E. hormaechei was found before in PIF, are the strains in this study related to those strains, do they contain the same virulence genes, is astA also found in other E. hormaechi? Was that stated before? Anything special about the K. cowanii strains irt what is already stated in PIF? 

The authors overstate the impact, first as only data from two countries is being tested and shown and second because the virulence of the strains is unclear (see other comments), and third because the prevalence is unclear (see above). The conclusions thus need to be specified to these regions or countries and can not be drawn in general and need to be put in context of virulence and prevalence (for instance statements in line 275 and line 267 and 289).

Other questions that can be answered are: do the AMR genes link to the phenotypic resistance found? 

Text need to be checked for italics of genes and species. And for typo's (in molecular - line 153)

Line 214 indicates Codex to recommend. this makes it unclear if it is a recommendation or a requirement to have no ENT.

line 50: why other ENT species are no less important needs more introduction. 

Author Response

Point 1: In this manuscript the authors describe what kind of bacterial contamination they have found in PIF from two countries, Mexico and Chile.

Although they have found contaminations, it is hard to determine the significance of the findings as it is unclear how many samples were tested to get to these numbers of contaminating strains. This needs to be included or at least discussed. Moreover it is also unclear if maybe some of the contaminations come from one sample.

Response Point 1: We analyzed 155 samples of which 80 came from Chile and 75 from Mexico. The prevalence of ENT was 8.4% for Chile and 3.8% for Mexico. All strains came from different samples. This was added in the manuscript.

Point 2: E. hormaechei was found before in PIF. Was K. cowanii found before in PIF? This needs to be stated to determine the novelty. Also for resistance genes found the novelty (for PIF) needs to be stated. In general it is unclear if the results presented are novel or a confirmation of what was already known. For instances, E. hormaechei was found before in PIF, are the strains in this study related to those strains, do they contain the same virulence genes, is astA also found in other E. hormaechi? Was that stated before? Anything special about the K. cowanii strains irt what is already stated in PIF?

Response Point 2 : Enterobacter hormaechei has been associated with outbreaks and cases related to PIF (Townsend et al., 2008; Parra-Flores et al., 2018) and nosocomial outbreaks (Roberts et al., 2020). This microorganism has been reported and mistaken for a false C. sakazakii because outdated identification methods have been used. Several cases of outbreaks associated with E. hormaechei have been reported, but which at the time were erroneously identified as caused by C. sakazakii (Caubilla-Barron et al. 2007; Townsend et al. 2008; Jackson et al. 2015). Therefore, reporting E. hormaechei in PIF distributed in a whole continent is a discovery, especially if this microorganism presents several virulence and resistance genes. It is a novelty of this manuscript.

Parra–Flores et al. (2020) published a study on PIF in which E. hormaechei and K. cowanii were found in PIF distributed in the whole continent of the Americas. These authors identified them by sequencing the fusA gene. Therefore, K.cowanii has only been reported once in PIF and sequencing its genome made it possible to show pathogenicity characteristics, given that it has been recently mentioned with clinical cases in humans.

As for the astA gene, we have not found it in other strains of E. hormaechei, and it has not been reported recently. The astA gene is frequently reported for E. coli and in a previous study (unpublished), we found the astA gene in E. kobei associated with a diarrhea outbreak in children.

Point 3: The authors overstate the impact, first as only data from two countries is being tested and shown and second because the virulence of the strains is unclear (see other comments), and third because the prevalence is unclear (see above). The conclusions thus need to be specified to these regions or countries and can not be drawn in general and need to be put in context of virulence and prevalence (for instance statements in line 275 and line 267 and 289).

Response Point 3: Regarding what was mentioned by the reviewer, we revised the wording so that the comments do not appear to be exaggerated and we circumscribed the findings associated with the places where these PIF are marketed. However, finding these microorganisms in PIF intended for consumption by newborns and distributed throughout the Americas indicates the lack of hygienic control of these products by the companies that manufacture them. 

Point 4: Other questions that can be answered are: do the AMR genes link to the phenotypic resistance found?

Response Point 4: There is a link between AMR and phenotypic resistance. For example, lines 231-236. The development of antibiotic resistance is usually associated with genetic changes, either to the acquisition of resistance genes, or to mutations in elements relevant for the activity of the antibiotic. Recent work has also shown that the susceptibility to antibiotics is highly dependent on the bacterial metabolism and that global metabolic regulators can modulate this phenotype. This modulation includes situations in which bacteria can be more resistant or more susceptible to antibiotics. Understanding these processes will thus help in establishing novel therapeutic approaches based on the actual susceptibility shown by bacteria during infection, which might differ from that determined in the laboratory (Corona and Martinez, 2013). Therefore, we wanted to show this link between phenotype and resistance genes in Table 3.  

Point 5: Text need to be checked for italics of genes and species. And for typo's (in molecular - line 153)

Response Point 5: This was revised and corrected.

Point 6: Line 214 indicates Codex to recommend. this makes it unclear if it is a recommendation or a requirement to have no ENT.

Response Point 6: The Codex Commission establishes recommendations because countries are those responsible to establish the control standards. The Codex Commission states: “The objective of this Code is to provide practical guidance and recommendations to governments, industry, health care professionals/caregivers of infants and young children, as appropriate, on the hygienic manufacture of PF and on the subsequent hygienic preparation, handling and use of reconstituted formulae. The Code identifies relevant control measures at the various steps in the food chain that can be employed to reduce the risks for infants and young children that are associated with the consumption of PF.” In addition, it establishes the verification indicators n = 10, C = 2, and m = 0/10 g.

Point 7: line 50: why other ENT species are no less important needs more introduction.

Response Point 7: The writing of the manuscript was improved and and we added: A recent Center for Disease Control (CDC) study that summarized data collected from 2011 to 2017 found that ENT caused between 23% and 31% of infections in adult, pediatric, and oncology wards. (The Changing Face of the Family Enterobacteriaceae (Order: “Enterobacterales”): New Members, Taxonomic Issues, Geographic Expansion, and New Diseases and Disease Syndromes.  Janda JM, Abbott SL. The Changing Face of the Family Enterobacteriaceae (Order: "Enterobacterales"): New Members, Taxonomic Issues, Geographic Expansion, and New Diseases and Disease Syndromes. Clin Microbiol Rev. 2021, 24;34(2):e00174-20. doi: 10.1128/CMR.00174-20).